# Crystal Evolution of Calcium Silicate Minerals Synthesized by Calcium Silicon Slag and Silica Fume with Increase of Hydrothermal Synthesis Temperature

**DOI:** 10.3390/ma15041620

**Published:** 2022-02-21

**Authors:** Zhijie Yang, De Zhang, Yang Jiao, Chengyang Fang, Dong Kang, Changwang Yan, Ju Zhang

**Affiliations:** 1School of Mining and Technology, Inner Mongolia University of Technology, Hohhot 010051, China; 20201100458@imut.edu.cn (D.Z.); 20211800659@imut.edu.cn (Y.J.); 20211100472@imut.edu.cn (C.F.); 20201100459@imut.edu.cn (D.K.); yanchangwang@imut.edu.cn (C.Y.); zj970741@imut.edu.cn (J.Z.); 2The Key Laboratory of Green Development for Mineral Resources, Inner Mongolia University of Technology, Hohhot 010051, China

**Keywords:** calcium silicon slag, silica fume, hydrothermal synthesis temperature, calcium silicate minerals, crystal evolution

## Abstract

In order to realize high-value utilization of calcium silicon slag (CSS) and silica fume (SF), the dynamic hydrothermal synthesis experiments of CSS and SF were carried out under different hydrothermal synthesis temperatures. In addition, phase category, microstructure, and micropore parameters of the synthesis product were analyzed through testing methods of XRD, SEM, EDS and micropore analysis. The results show that the main mechanism of synthesis reaction is that firstly β-Dicalcium silicate, the main mineral in CSS, hydrates to produce amorphous C–S–H and Ca(OH)_2_, and the environment of system is induced to strong alkaline. Therefore, the highly polymerized Si-O bond of SF is broken under the polarization of OH^−^ to form (SiO_4_) of Q^0^. Next, amorphous C–S–H, Ca(OH)_2_ and (SiO_4_) of Q^0^ react each other to gradually produce various of calcium silicate minerals. With an increase of synthesis temperature, the crystal evolution order for calcium silicate minerals is cocoon-like C–S–H, mesh-like C–S–H, large flake-like gyrolite, small flake-like gyrolite, petal-like gyrolite, square flake-like calcium silicate hydroxide hydrate, and strip-like tobermorite. In addition, petal-like calcium silicate with high average pore volume (APV), specific surface area (SSA) and low average pore diameter (APD) can be prepared under the 230 °C synthesis condition.

## 1. Introduction

Calcium silicate hydrate minerals based on CaO-SiO_2_-H_2_O system can be prepared using different calcium raw materials and silicate raw materials under different hydrothermal synthesis conditions. These crystal calcium silicate minerals take on different microstructure such as agglomerate-like [1], honeycomb-like [2], rod-like [3], fibrous-like [4], cocoon-like [5], needle-like [6] and so on. At the same time, these different crystal of calcium silicate minerals can be used in papermaking [7], rubber [8], building materials [9,10], sewage treatment and air purification [11,12,13,14], thermal insulation material [15] and other fields. The CSS is as a new type of waste discharged from aluminum extraction technology using fly ash [16], and its main chemical components are CaO and SiO_2_. Therefore, the authors carried out preparation of different calcium silicate minerals using CSS and SF as the main raw materials through the dynamic hydrothermal synthesis process [17]. It not only solved the problem of CSS treatment that restricts the development of aluminum extraction from fly ash, but also realized the high-value utilization of CSS. However, in the process of promoting industrial application of the synthesized calcium silicate minerals, it is found that the quality of different crystal calcium silicate minerals fluctuates greatly, which is difficult to meet the requirements of the downstream industry. The main reason is that the hydrothermal synthesis mechanism of calcium silicate minerals synthesized by CSS and SF is not clear. Furthermore, it is found that synthesis temperature is one of key factors affecting the crystal of calcium silicate minerals. Wei Guan synthesized porous calcium silicate mineral with highly active SiO_2_ at 110 °C through hydrothermal synthesis [18]. Ezgi Ogur synthesized xonotlite as the main phase using waste glass and lime, when synthesis temperature was at 220 °C and C/S molar ratio was 0.83 [19], but there is little study on how the synthesis temperature affects crystal of calcium silicate minerals synthesized by CSS and SF. 

Therefore, in order to clarify crystal control theory of calcium silicate minerals, in the paper, the hydrothermal synthesis experiments of calcium silicate minerals using CSS and SF were conducted under different synthesis temperature. In addition, the hydrothermal synthesis reaction mechanism of CSS and SF was discussed to clarify the effect of synthesis temperature on the formation and evolution of calcium silicate minerals with different crystal.

## 2. Materials and Methods

### 2.1. Materials

The CSS, SF and distilled water are used as main raw materials in experiment. The CSS was supplied by Datang Renewable Resources company, which built first production line of extracting aluminum from fly ash in the world. The SF was supplied by Inner Mongolia Erdos Ferrosilicon alloy company. The chemical and phase composition of CCS and SF are shown in Table 1 and Figure 1. The main phases of CSS are β-dicalcium silicate and calcite. The main phase of SF is glass phase.

### 2.2. Methods

The CSS was dried in blast drying oven at 110 °C for about 5 h, and then CSS dried was milled to particle size less than 45 μm in a ball mill again. According to the 100 g CSS and 58.23 g SF were weighed and 2373.45 g distilled water was added to make C/S molar ratio of 0.9 and liquid-solid ratio of 15:1 for each synthesis experiment. After mixing, the mixture was put into a 5 L high-pressure reactor. The dynamic hydrothermal synthesis experiments were conducted according to process parameters of the stirring rate 300 r/min and the heating rate 8 °C/min to corresponding synthesis temperature and constant temperature for 6 h. Then when high-pressure reactor was gradually cooled to about 20 °C, the synthetic material was removed from the high-pressure reactor and dehydrated through a filter and dried in a blast drying oven at 110 °C for 5 h. 

Hydrothermal synthesis temperature from 180 °C to 260 °C, one synthesis experiment was executed for every 10 °C, a total of 9 experiments. Finally, XRD (PANalytical, Almelo, The Netherlands, X’Pert Powder 3, Cu target, 40 kV, step 0.02°), SEM (S-4800, Hitachi, Tokyo, Japan) and micropore analyzer (V-Sorrb 2800TP, Gold APP Instruments Corporation, Beijing, China) were used to analyze the phase category, micromorphology, micropore parameters of synthesis product. 

## 3. Results and Discussion

### 3.1. Phase Evolution of Calcium Silicate Minerals with Increase of Hydrothermal Synthesis Temperature

XRD analysis result of synthesis products at different synthesis temperatures is shown in Figure 2, and mineral phase information of synthesis calcium silicate is also shown in Table 2. When the synthesis temperature is between 180 °C and 200 °C, the main phases in the synthesis products are β-dicalcium silicate, calcite, and crystalline C–S–H. That because the β-dicalcium silicate, main phase in the CSS, occurs hydration reaction according to Equation (1) under dynamic hydrothermal synthesis conditions. Furthermore, amorphous C–S–H and Ca(OH)_2_ are generated at first. Then under high temperature and high pressure, amorphous C–S–H is further crystallized to form crystalline C–S–H. Meanwhile, the environment of system is induced to strong alkaline. Therefore, the highly polymerized Si-O bond of SF are broken under the polarization of OH^−^ to form (SiO_4_) of Q^0^ [20], and reacts with Ca(OH)_2_ to form crystalline C–S–H, and this mechanism was also confirmed by Rachel Camerini in studying the synthesis kinetics of calcium silicate hydrate [21]. However, because β-dicalcium silicate is a slower hydration rate mineral and its hydration degree is less than 10.3% in 28 days [22], β-dicalcium silicate in CSS cannot be completely hydrated and will be partially remained. At the same time, because calcite almost does not hydrolyze or react with SF under condition of high temperature and pressure, the calcite phase contained in synthesis products may be derived from the original calcite in the CSS.
(1)$$2CaO\cdot SiO_{2}+mH_{2}O=xCaO\cdot SiO_{2}\cdot \left(m+x-2\right)H_{2}O+\left(2-x\right)Ca{\left(OH\right)}_{2}$$

Since β-dicalcium silicate hydration will be intensified with temperature increase. Therefore, when the synthesis temperature increases to 210–230 °C, the more Ca(OH)_2_ is generated. Therefore, the reaction between Ca(OH)_2_ with SF is intensified, and generating gyrolite (Ca_4_(Si_6_O_15_)(OH)_2_·3H_2_O) phase with higher C/S molar ratio. When the synthesis temperature increased to 240 °C, not only Calcium silicate hydroxide hydrate (Ca_4.5_Si_6_O_15_(OH)_2_·3H_2_O) and tobermorite (5CaO·6SiO_2_·5H_2_O) appear in the synthesis products, but also the phases of β-dicalcium silicate and crystalline C–S–H disappears. This indicates that under synthesis temperature of 240 °C and synthesis time of 6 h, β-dicalcium silicate has been completely hydrated, and a large number of Ca(OH)_2_ reacted with SF to generate calcium silicate minerals with higher C/S molar ratio. This is basically similar to the experimental results of synthesis tobermorite with fly ash and quartz or lime [23,24].

When the synthesis temperature continues to rise to 250 °C, there are only tobermorite, calcium silicate hydroxide hydrate and calcite phase in the synthesis products, while the gyrolite phase disappears. When the synthesis temperature reaches 260 °C, calcium silicate hydroxide hydrate phase also disappears, only tobermorite and calcite phase are remained. This shows that, with increase of synthesis temperature, the hydrothermal synthesis reaction of CSS and SF will generate calcium silicate mineral with higher and higher C/S molar ratio. 

Based on the above analysis, the main synthesis reaction mechanism is that firstly β-dicalcium silicate, the main mineral in CSS, hydrates to produce amorphous C–S–H and Ca(OH)_2_, and inducing strong alkaline environment in system. Therefore, the highly polymerized Si–O bond of SF are broken under the polarization of OH^−^ to form (SiO_4_) of Q^0^. Then at condition of high temperature and high press, amorphous C–S–H, Ca(OH)_2_ and (SiO_4_) of Q^0^ react each other to gradually produce various of calcium silicate minerals with higher and higher C/S molar ratio with increase of synthesis temperature.

### 3.2. Micromorphology Evolution of Calcium Silicate Minerals with Increase of Hydrothermal Synthesis Temperature

SEM images of 50,000 times and 10,000 times are shown as follows in Figure 3. 

In addition, atom molar ratio of special regions was tested by EDS to infer the phase of calcium silicate minerals with different micromorphology, and the final results are shown in Table 3.

Figure 3 shows that the micromorphology of main synthesis products is cocoon-like sphere at temperatures of 180 °C, as shown in Figure 3A. The average particle size of cocoon-like sphere crystal is about 1.5 mm, and the EDS analysis result is shown in region 1 in Table 3. However, in addition to cocoon-like crystalline C–S–H, there are other block-like minerals. EDS analysis results indicate block-like minerals are calcite as shown region 2 of Table 3. The atom molar ratio of cocoon-like phase is similar to crystalline C–S–H, which can be inferred as crystalline C–S–H phase. When the synthesis temperature increases to 190 °C and 200 °C, the micromorphology of synthesis product takes on mesh-like, as shown in Figure 3B,C. The EDS analysis result are shown in region 3 and region 5 of Table 3. The atom molar ratio of mesh-like phase is also similar to crystalline C–S–H, which can be inferred as crystalline C–S–H phase. However, there are a large number of granular-like minerals in Figure 3b, and the EDS results show the granular-like minerals are β-dicalcium silicate as shown in region 4 of Table 3. Furthermore, the block-like minerals are verified as calicate as as shown in region 6 of Table 3. Therefore, although the main phase of synthesis products is crystalline C–S–H at temperature 180 °C, 190 °C and 200 °C, their crystal is different. Therefore, combining with the XRD analysis results, it can be inferred that the amorphous C–S–H is firstly generated in the process of the dynamic hydrothermal synthesis, and it continues to develop and grow with the increase of synthesis temperature, and its crystal micromorphology evolves from cocoon-like to mesh-like. At the same time, there are also incomplete hydration β-dicalcium silicate and calcite generated by carbonization, which is also consistent with the XRD analysis above. 

When the synthesis temperature rises to 210 °C, it is found in Figure 3D,d that large flake-like phase grows into the mesh-like phase. The EDS analysis results are shown in region 7 of Table 3. The atom molar ratio of these large flake-like phase is similar to the gyrolite, which can be inferred as the gyrolite phase [25]. In addition, mesh-like minerals are still crystalline C–S–H as shown in region 8 of Table 3. When the synthesis temperature rises to 220 °C, the micromorphology of synthesis products is also flake-like, but these flake-like phases intertwine each other, as shown in Figure 3E. Compared with the flake-like phase generated at 210 °C, the grain size is smaller. The EDS analysis result is shown in region 9 of Table 3. The atom molar ratio of small flake-like phase is similar to the gyrolite, so small flake-like phase is the gyrolite. However, the Figure 3e shows that there is still incompletely transformed crystalline C–S–H at hydrothermal synthesis of 220 °C, as shown in region 10 of Table 3. When the synthesis temperature continues to rise to 230 °C, it is observed in Figure 3F that the micromorphology of the synthesis product is petal-like, and the connection of each flake-like grain is closer, and the grain size is smaller. EDS analysis results are shown in region 11 of Table 3. The atom molar ratio of these petal-like phases is similar to the gyrolite, which can be inferred that it is also the gyrolite. At the same time, it can be seen from Figure 3f and region 12 of Table 3 that a small number of small flake-like gyrolite has not transformed into petal-like minerals. Combined with the results of XRD analysis, it can be speculated that when the synthesis temperature rises to 210 °C, the mesh-like crystalline C–S–H reacts with Ca(OH)_2_ and (SiO_4_) of Q^0^ to form large flake-like gyrolite. Furthermore, with the increase of synthesis temperature, more flake-like gyrolite phase is generated. So, the crystal micromorphology evolves from large flake-like to small flake-like and finally to petal-like.

When the synthesis temperature increases to 240 °C and 250 °C, it is found from Figure 3G,H that the micromorphology of synthesis products is square flake-like, and these square flake-like phases are stacked together. EDS analysis results are shown in region 13 and region 15 of Table 3. The atom molar ratio of square flake phase is similar to calcium silicate hydroxide hydrate, which can be inferred as calcium silicate hydroxide hydrate. Combined with XRD analysis results, it can be speculated that when the synthesis temperature rises to 240 °C, the square flake-like calcium silicate hydroxide hydrate is generated through reaction of flake-like gyrolite and Ca(OH)_2_. Therefore, crystal micromorphology evolves from petal-like to square flake-like at 230 °C. In addition, it is found from Figure 3g,h that there is a small amount of strip-like minerals and EDS analysis results show these strip-like minerals are tobermorite as shown in region 14 and region 16 of Table 3. Therefore, it will further confirm that tobermorite is gradually formed when sythesis temperature reaches more than 240 °C. 

When the synthesis temperature continues to rise to 260 °C, it is found from Figure 3I,i that the micromorphology of the synthesis products is strip-like, and EDS analysis result is shown in the region 17 and region 18 of Table 3. Combined with XRD analysis results, it can be speculated that the atom molar ratio of strip-like phase is similar to tobermorite, which can be inferred as tobermorite phase. Therefore, it indicates, when the synthesis temperature rises to 260 °C, the square flake-like calcium silicate hydroxide hydrate reacted with Ca(OH)_2_ and (SiO_4_) of Q^0^ to form strip-like tobermorite. Meanwhile, it can be seen from Figure 3I,i that almost all synthesis products of CSS and SF are strip-like tobermorite. 

According to the above analysis, it can be seen that with increase of synthesis temperature, the main crystal evolution order of calcium silicate minerals synthesized by CSS and SF is cocoon-like C–S–H, mesh-like C–S–H, large flake-like gyrolite, small flake-like gyrolite, petal-like gyrolite, square flake-like galcium silicate hydroxide hydrate, and strip-like tobermorite.

### 3.3. Micropore Parameters Evolution of Calcium Silicate Minerals with Increase of Hydrothermal Synthesis Temperature

The micropore analyzer was used to analyze the micropore parameters of synthesis products of CSS and SF at different synthesis temperatures, and the results are shown in Figure 4. The APV, APD and SSA of synthesis products increase first and then decrease with increase of synthesis temperature.

According to Figure 4 and results of XRD and SEM, between synthesis temperature 180 °C and 200 °C, the APV, APD and SSA of the synthesis products are relatively lower. At this synthesis temperature, the main phase is crystalline C–S–H, with the increase of the synthesis temperature, the crystal evolves from cocoon-like crystalline C–S–H to mesh-like crystalline C–S–H, and the grain size and pore size increase significantly. When the temperature rises to 210 °C and 230 °C, the crystal of synthesis products evolves from crystalline C–S–H to flake-like gyrolite, and its SSA and APD increase significantly. Especially when the synthesis temperature is at 210 °C, the APV, APD and SSA are very large due to the formation of large flake-like gyrolite. However, when the temperature is at 230 °C, because the petal-like gyrolite is semi-closed pores, compared with the flake-like structure formed at 210 °C and 220 °C, the APV and SSA of the synthesis products are relatively higher, but APD is relatively lower, as shown in Figure 4. In addition, there are also corresponding experimental results in the study of pore structure of calcium silicate hydrate [26,27]. When the temperature rises to 240 °C, 250 °C, 260 °C, APV, APD and SSA are lower, because crystal of main synthesis products is square flake-like and strip-like.

According to the above analysis results, with increase of synthesis temperature, the APV, APD and SSA of the synthesis products synthesized by CSS and SF show a change trend of first increase and then decrease. The average pore volume, average pore diameter and specific surface area are closely related to the crystal of main synthesis products. The APV, APD and SSA are smaller, when main synthesis products are cocoon-like and mesh-like crystalline C–S–H, square flake-like calcium silicate hydroxide hydrate and strip-like tobermorite. While the APV, APD and SSA are larger, when main synthesis products are flake-like gyrolite. However, because the petal-like gyrolite is semi-closed pores, it’s the APV and SSA are relatively higher, but the APD is relatively lower.

## 4. Conclusions

In the paper, the dynamic hydrothermal synthesis experiments of CSS and SF were conducted under different hydrothermal synthesis temperature, and the phase category, microstructure, micropore parameters of the synthesis products were analyzed through the different testing methods, such as XRD, SEM, EDS, the micropore analysis and so on. The main aim is to discuss the hydrothermal synthesis reaction mechanism of CSS and SF and clarify the effect of synthesis temperature on the crystal formation and evolution of calcium silicate minerals. The conclusions are shown as follows:
(1)The main reaction mechanism of hydrothermal synthesis is that β-dicalcium silicate contained in CSS, firstly hydrates to generate amorphous C–S–H and Ca(OH)_2_, and the environment of system is induced to strong alkaline. Therefore, the highly polymerized Si–O bond of SF are broken under the polarization of OH^−^ to form (SiO_4_) of Q^0^. At condition of high temperature and high press, amorphous C–S–H, Ca(OH)_2_ and (SiO_4_) of Q^0^ react each other to gradually produce various of calcium silicate minerals with higher and higher C/S molar ratio with increase of synthesis temperature.(2)With the increase of synthesis temperature, crystal evolution order of calcium silicate mineral synthesized by CSS and SF is cocoon-like C–S–H, mesh-like C–S–H, large flake-like gyrolite, small flake-like gyrolite, petal-like gyrolite, square flake-like calcium silicate hydroxide hydrate, and strip-like tobermorite.(3)With the increase of synthesis temperature, the APV, APD and SSA of the synthesis products show a change trend of first increase and then decrease. The APV, APD and SSA are closely related to the main crystal of synthesis products. However, because petal-like gyrolite is semi-closed pores, it’s APV and SSA are relatively higher, but it’s APD is relatively lower.

## Figures and Tables

**Figure 1 materials-15-01620-f001:**
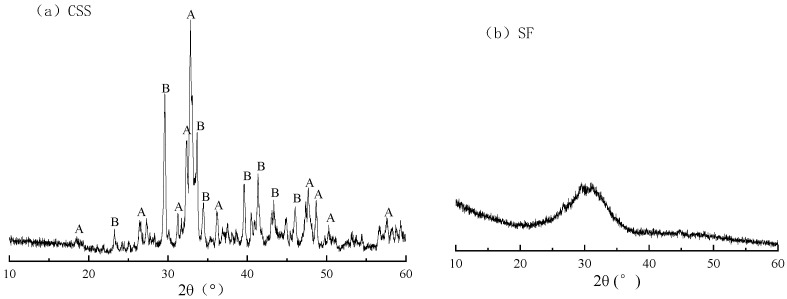
XRD patterns of CSS and SF; (**a**)-β-dicalcium silicate (β-2CaO·SiO_2_) (**b**)-calcite (CaCO_3_).

**Figure 2 materials-15-01620-f002:**
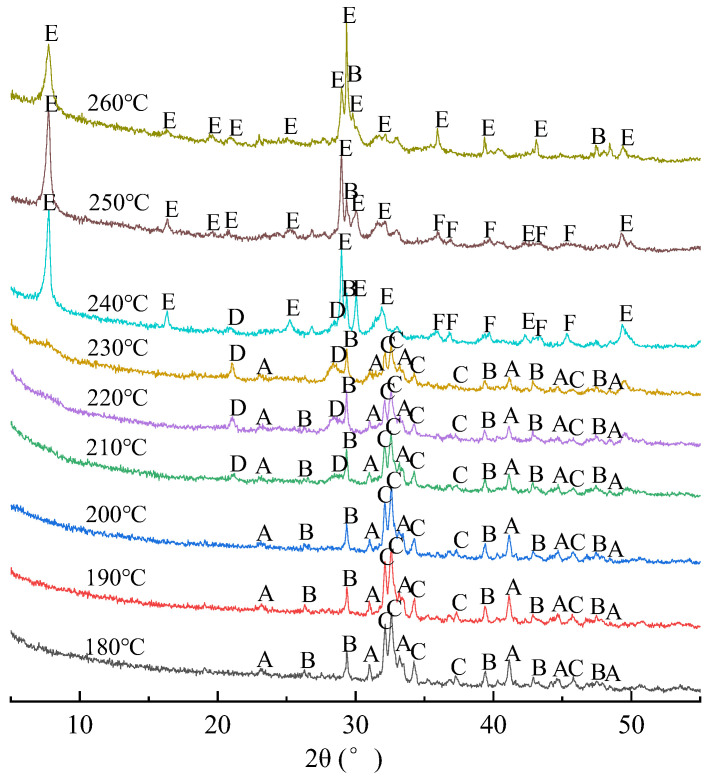
XRD patterns of synthesis products. A—dicalcium silicate (β-2CaO·SiO_2_), B—calcite (CaCO_3_), C—crystalline C–S–H, D—gyrolite (Ca_4_(Si_6_O_15_)(OH)_2_·3H_2_O), E—tobermorite (5CaO·6SiO_2_·5H_2_O), F—calcium silicate hydroxide hydrate (Ca_4.5_Si_6_O_15_(OH)_2_·3H_2_O).

**Figure 3 materials-15-01620-f003:**
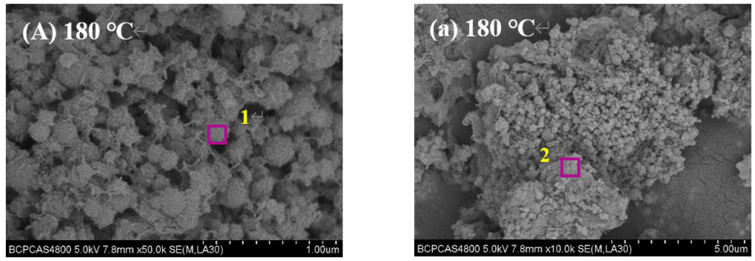
SEM images of synthesis products under different hydrothermal synthesis temperature; the uppercase letters represent SEM images of 50,000 times while the lowercase letters represent SEM images of 10,000 times.

**Figure 4 materials-15-01620-f004:**
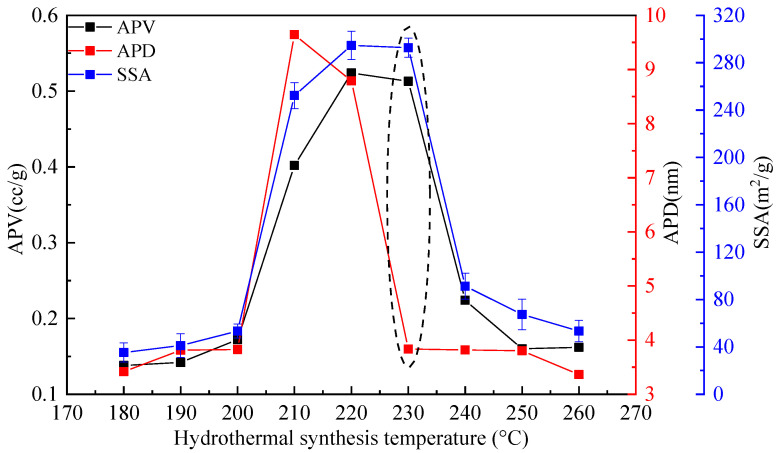
Change of micropore parameters for the synthesis product with increase of hydrothermal synthesis temperature.

**Table 1 materials-15-01620-t001:** Content of chemical composition for raw materials (calculated by mass fraction %).

Chemical Composition	SiO_2_	Fe_2_O_3_	Al_2_O_3_	CaO	MgO	Na_2_O	K_2_O	SO_3_	P_2_O_5_	F	Cl
CSS	29.26	2.53	5.30	55.50	3.61	2.58	0.36	0.73	0.14	-	0.42
SF	72.15	1.16	0.59	7.50	7.43	0.68	4.20	3.17	0.56	1.74	0.82

**Table 2 materials-15-01620-t002:** Minerals phase information of hydrothermal synthesis of hydrated calcium silicate.

No.	Phase	Chemical Formula	PDF Card No.	Main 2θ (°)
A	β-dicalcium silicate	β-2CaO·SiO_2_	01-083-0460	23.190, 32.169, 41.174
B	calcite	CaCO_3_	01-072-1937	29.369, 39.370, 48.452
C	crystalline C–S–H	C–S–H	00-002-0068	30.168, 31.589, 37.281
D	gyrolite	Ca_4_(Si_6_O_15_)(OH)_2_·3H_2_O	00-042-1425	21.065, 28.227, 31.797
E	tobermorite	5CaO·6SiO_2_·5H_2_O	00-045-1480	16.251, 30.044, 31.867
F	calcium silicate hydroxide hydrate	Ca_4.5_Si_6_O_15_(OH)_2_·3H_2_O	00-043-1488	35.817, 36.782, 45.291

**Table 3 materials-15-01620-t003:** EDS analysis results for different microcosmic region.

Microcosmic Region	Micromorphology	Atom Molar Ratio (%)								Corresponding Phase
		O	Al	Si	Ca	Na	Mg	Fe	C	
1	cocoon-like	63.54	0.52	17.15	16.42	0.24	0.81	1.32	-	crystalline C–S–H
2	block-like	58.72	0.21	0.10	20.04	-	0.10	0.10	20.73	calcite
3	mesh-like	63.36	0.52	17.22	16.72	0.35	0.81	1.02	-	crystalline C–S–H
4	granular-like	54.42	1.75	14.76	27.02	0.56	1.13	0.36	-	β-dicalcium silicate
5	mesh-like	65.72	0.33	16.95	16.23	0.12	0.44	0.21	-	crystalline C–S–H
6	block-like	57.72	0.53	0.21	20.46	-	0.21	0.24	20.63	calcite
7	large flake-like	67.32	0.15	19.25	12.42	0.21	0.52	0.13	-	gyrolite
8	mesh-like	66.52	0.33	16.15	16.23	0.12	0.44	0.21	-	crystalline C–S–H
9	small flake-like	64.25	0.44	19.05	12.34	0.13	2.23	1.56	-	gyrolite
10	mesh-like	65.72	0.33	16.95	16.23	0.12	0.44	0.21	-	crystalline C–S–H
11	petal-like	67.44	0.12	20.35	11.03	0.65	0.36	0.05	-	gyrolite
12	small flake-like	64.15	0.44	18.45	12.64	0.23	2.23	1.56	-	gyrolite
13	square flake-like	64.07	0.05	20.67	13.41	0.34	1.27	0.19	-	calcium silicate hydroxide hydrate
14	strip-like	66.22	0.33	17.45	15.23	0.12	0.44	0.21	-	tobermorite
15	square flake-like	65.27	0.19	19.47	13.09	0.27	1.66	0.05	-	calcium silicate hydroxide hydrate
16	strip-like	64.67	0.36	18.35	15.42	0.45	0.52	0.23	-	tobermorite
17	strip-like	65.34	0.34	17.28	15.37	0.26	0.41	1.00	-	tobermorite
18	strip-like	66.23	0.26	17.91	14.83	0.14	0.32	0.31	-	tobermorite

## Data Availability

The data presented in this study are available on request from the corresponding author. At the time the project was carried out, there was no obligation to make the data publicly available.

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
