# Peer review of "Crystal Evolution of Calcium Silicate Minerals Synthesized by Calcium Silicon Slag and Silica Fume with Increase of Hydrothermal Synthesis Temperature"

_materials, 2022, doi:10.3390/ma15041620_

Round 1

Reviewer 1 Report

1. Abbreviations must be defined at their first mention in a manuscript. The abbreviations, CSS and SF, may not be used in title. 

2. Why does tobermorite only exist at 240°C and above? Tobermorite should be exit at the lower temperatures.

3. In Fig. 4, the changes in pore structures (APV, APD, and SSA) seem strange, and there are no references supporting the findings. The authors shoud provide more experimantal results for explanation.

4. What is the applicaitons of the calcium silicate mineral synthesized in this study?

5. This manuscript has few references for discussion. The authors must add some references to compare the results with other studies and to support the findings in this research.

6. Overall, the data in this manuscript are insufficient, and the organization, innovation, and academic contribution should be improved.  

Author Response

1. Abbreviations must be defined at their first mention in a manuscript. The abbreviations, CSS and SF, may not be used in title. 

Response: The CSS and SF in title has been modified to calcium silicon slag and silica fume, respectively.

2. Why does tobermorite only exist at 240°C and above? Tobermorite should be exit at the lower temperatures.

Response: The theoretical C/S molar ratio of tobermorite is 0.83, while the raw material C/S molar ratio set in this study is 0.9. The calcium source for the synthesis of tobermorite mainly comes from β-Dicalcium silicate contained in calcium silicate slag. But β- dicalcium silicate is a slow hydration rate mineral, only when the temperature rises to 240 ℃,β- dicalcium silicate can be fully hydrated. So tobermorite can be formed only when synthesis temperature is above 240 ℃.

3. In Fig. 4, the changes in pore structures (APV, APD, and SSA) seem strange, and there are no references supporting the findings. The authors should provide more experimental results for explanation.

Response: The test results of micropore parameters of calcium silicate minerals are indeed shown as Figure. 4. The variation law of APV, APD and SSA with synthesis temperature is mainly determined by the micromorphology of the synthesis product. And calcium silicate slag is as a new special solid waste, there is little research on it. Especially in the hydrothermal synthesis of calcium silicate minerals by using calcium silicate slag, there are almost no relevant reports except for the author's research. Therefore, it is difficult to find relevant references for comparison.

4. What is the applicaitons of the calcium silicate mineral synthesized in this study?

Response: The synthetic petal-like gyrolite is mainly used as plastic filler, and the synthetic strip-like tobermorite is mainly used to prepare calcium silicate thermal insulation material.

5. This manuscript has few references for discussion. The authors must add some references to compare the results with other studies and to support the findings in this research.

Response: More references are added and discussed in the manuscript, and the details are shown in References.

6. Overall, the data in this manuscript are insufficient, and the organization, innovation, and academic contribution should be improved.  

Response: In order to further improve the research content, the PDF card number for each detected phase has been shown in Table 2 of manuscript. And SEM images of 10000 times and EDS results of other phase have been supplemented and explained in manuscript, and the details are shown in Figure.3 and Table 3. At the same time, the novelty of this paper is highlighted through the optimization of the full text.

Reviewer 2 Report

Title:

The paper discusses the hydrothermal synthesis of calcium silicate minerals through the interaction of CSS and SF. English should be fully revised and polished. However, much of the data presented in this manuscript appears similar to those in another publication, as in the following citation link:

Zhijie Yang, Dong Kang, De Zhang, Changwang Yan, Ju Zhang, Crystal transformation of calcium silicate minerals synthesized by calcium silicate slag and silica fume with increase of C/S molar ratio, Journal of Materials Research and Technology,

Volume 15, 2021, Pages 4185-4192, https://doi.org/10.1016/j.jmrt.2021.10.047.

The paper is accordingly rejected. The following comments are provided to improve the quality of the paper after removing the similarity with the above-mentioned work.

Please spell out the abbreviations (CSS, SF, APV, SSA, APD, etc.)

L12: temperature: temperatures

L12: And the phase category: The phase category,

L12: micropore parameters of the synthesis: and micropore parameters of the synthesis

L16: is induced strong alkaline: is induced to be strongly alkaline

L21-L22: Petal-like calcium silicate with high APV, SSA, and low APD can also be prepared under 230 °C synthesis conditions.

L27-L3: too long sentence

L30-L32: reference (7) is missing.

L36: minerals,

L38: high-value….

L38: extraction aluminum: aluminum extraction

L51-L56: too long sentence!

L51: theoretical of calcium silicate mineral: theory of calcium silicate mineral

L55-L56: different crystals.

L68-L71: should come before Table 1 and Figure 1.

L74: size less than 300 mesh: it better to express the size in micron.

L131-L133: this part is repeated from the abstract.

L143: as follows in Figure 3

L152: at temperatures of 180 °C

their crystals are different.

L173: And the EDS analysis…. Remove “And” from the manuscript.

L180:  When the synthesis temperature…(remove space).

L249-L250: The main aim is to discuss the reaction mechanism of hydrothermal synthesis of CSS and SF?

It is the reaction of CSS and SF, not the synthesis of CSS and SF

Author Response

1. The paper discusses the hydrothermal synthesis of calcium silicate minerals through the interaction of CSS and SF. English should be fully revised and polished. However, much of the data presented in this manuscript appears similar to those in another publication, as in the following citation link:

Zhijie Yang, Dong Kang, De Zhang, Changwang Yan, Ju Zhang, Crystal transformation of calcium silicate minerals synthesized by calcium silicate slag and silica fume with increase of C/S molar ratio, Journal of Materials Research and Technology, Volume 15, 2021, Pages 4185-4192, https://doi.org/10.1016/j.jmrt.2021.10.047.

Response: The main content of my previous article, published in Journal of Materials research and technology 2021; 1 5: 4185 -4192, is about crystal transformation of calcium silicate minerals with increase of C/S molar ratio. It is a sister article to this manuscript, which main study content is about Crystal evolution of calcium silicate mineral with increase of hydrothermal synthesis temperature.

2. The paper is accordingly rejected. The following comments are provided to improve the quality of the paper after removing the similarity with the above-mentioned work.

Please spell out the abbreviations (CSS, SF, APV, SSA, APD, etc.)

L12: temperature: temperatures

L12: And the phase category: The phase category,

L12: micropore parameters of the synthesis: and micropore parameters of the synthesis

L16: is induced strong alkaline: is induced to be strongly alkaline

L21-L22: Petal-like calcium silicate with high APV, SSA, and low APD can also be prepared under 230 °C synthesis conditions.

L27-L3: too long sentence

L30-L32: reference (7) is missing.

L36: minerals,

L38: high-value….

L38: extraction aluminum: aluminum extraction

L51-L56: too long sentence!

L51: theoretical of calcium silicate mineral: theory of calcium silicate mineral

L55-L56: different crystals.

L68-L71: should come before Table 1 and Figure 1.

L74: size less than 300 mesh: it better to express the size in micron.

L131-L133: this part is repeated from the abstract.

L143: as follows in Figure 3

L152: at temperatures of 180 °C

their crystals are different.

L173: And the EDS analysis…. Remove “And” from the manuscript.

L180:  When the synthesis temperature…(remove space).

L249-L250: The main aim is to discuss the reaction mechanism of hydrothermal synthesis of CSS and SF?

It is the reaction of CSS and SF, not the synthesis of CSS and SF

Response: CSS- Calcium silicon slag,SF- Silica fume, APV- Average pore volume, SSA- Specific surface area, APD- Average pore diameter.

The manuscript has been revised according to the reviewer's suggestions of L12-L250.

Reviewer 3 Report

This paper describes systematic results of the characterization of the reaction process between calcium silicate slag and silica fume. The contents are useful for readers concerning the utilization of calcium silicate slag for various applications. This paper is worthy of publication in Materials.

Author Response

Thank you very much for your valuable comments to this manuscript.

Reviewer 4 Report

The authors synthesized calcium silicate minerals via the hydrothermal method using CSS and SF as main raw materials. The phase, microstructure, and micropore parameters of the obtained materials were dependent on the synthesis temperature. The work is similar to previous authors' research, Journal of Materials research and technology 2021;1 5: 4185 -4192. Moreover, there are some problems in the writing and experimental analysis, so I suggest that it can be accepted after careful revision. 

  1. The authors should define the novelty of their work. 
  2. The PDF card number for each detected phase on diffractograms should be provided by the authors. 
  3. The XRD results indicate to multiphase of samples, meanwhile, the EDS results are related only to the one (main) phase. The authors should provide the EDS results from other sample regions and compare them to the XRD results. 
  4. The authors should provide the SEM images from the larger and smaller samples area to better present their morphology.
  5. The reaction mechanism of hydrothermal synthesis is unclear, the authors did not provide sufficient evidence to support these views and the logic of the article was not clear enough in the article.

Author Response

1. The authors should define the novelty of their work. 

Response:

1)Calcium silicate slag generated in process of alumina extraction from high-alumina fly ash, as a new type of solid waste, is first as a raw material to hydrothermal synthesis calcium silicate minerals.

2)Exploring the hydrothermal synthesis reaction mechanism of calcium silicate minerals synthesized by calcium silicate slag and silica fume.

3)Clarifying the crystal evolution law of calcium silicate minerals with the increase of hydrothermal synthesis temperature.

4)Realizing accurately control of crystal of calcium silicate minerals, and it will make petal-like gyrolite to be used as plastic filler, and strip-like tobermorite to be prepared calcium silicate thermal insulation material.

2. The PDF card number for each detected phase on diffractograms should be provided by the authors. 

Response: The PDF card number for each detected phase has been shown in Table 2 of manuscript.

3. The XRD results indicate to multiphase of samples, meanwhile, the EDS results are related only to the one (main) phase. The authors should provide the EDS results from other sample regions and compare them to the XRD results. 

Response: SEM images of 10000 times and EDS results of other phase have been supplemented and explained in manuscript. And the details are shown in Figure.3 and Table 3.

4. The authors should provide the SEM images from the larger and smaller samples area to better present their morphology.

Response: SEM images of 10000 times have been supplemented and explained in manuscript. And the details are shown in Figure.3 and Table 3.

5. The reaction mechanism of hydrothermal synthesis is unclear, the authors did not provide sufficient evidence to support these views and the logic of the article was not clear enough in the article.

Response: The reaction mechanism of hydrothermal synthesis has been shown in Conclusions (1) through analysis of XRD, SEM and EDS. The main reaction mechanism of hydrothermal synthesis is that β-dicalcium silicate contained in CSS, firstly hydrates to generate amorphous C-S-H and Ca (OH)2, and the environment of system is induced to strong alkaline. Then the Si-O bond of SF are broken and continue to react with the amorphous C-S-H and Ca (OH)2 to generate different crystal of calcium silicate minerals with the increase of synthesis temperature.

Round 2

Reviewer 1 Report

The authors did not well respond to the comments given before. The abbreviations are still wrong and the organization of this manuscript need to be improved.

Author Response

The authors did not well respond to the comments given before. The abbreviations are still wrong and the organization of this manuscript need to be improved.

Response: Thank you very much for your valuable comments to this manuscript again. I also read your comments very carefully. The CSS and SF in title has been modified to calcium silicon slag and silica fume, respectively. While, calcium silicon slag and silica fume are firstly abbreviated CSS and SF, when they firstly appear in first sentence of abstract. And average pore volume, average pore diameter and specific surface area are firstly abbreviated APV, APD and SSA, when they firstly appear in last sentence of abstract. Then their abbreviations have been used throughout following text. At the same time, the written expression and organization of whole manuscript also carefully revised and optimized again.

Reviewer 2 Report

Neglected recommendations:

L51-L56: too long sentence!

L68-L71: should come before Table 1 and Figure 1.

L129-L134: Too long sentence

This indicates that when the synthesis temperature is at 240 °C and synthesis time is 6 h, β-dicalcium silicate has been completely hydrated, and a large number of Ca(OH)2 reacted with SF to generate calcium silicate mineral with higher C/S molar ratio, which is basically similar to the experimental results of synthesis tobermorite with fly ash and quartz or lime [23, 24].

Please remove unnecessary "And"

Rephrase and split any long sentences!

L148: with higher and higher C/S moral ratio with increase of synthesis temperature.

It is C/S molar ratio

Please revise the whole manuscript again!

The reviewer cannot recommend this work for publication at this stage until the whole manuscript is carefully revised. The reviewer will leave the decision to the respected editor.

Author Response

L51-L56: too long sentence!

L68-L71: should come before Table 1 and Figure 1.

L129-L134: Too long sentence

This indicates that when the synthesis temperature is at 240 °C and synthesis time is 6 h, β-dicalcium silicate has been completely hydrated, and a large number of Ca(OH)2 reacted with SF to generate calcium silicate mineral with higher C/S molar ratio, which is basically similar to the experimental results of synthesis tobermorite with fly ash and quartz or lime [23, 24].

Please remove unnecessary "And"

Rephrase and split any long sentences!

L148: with higher and higher C/S moral ratio with increase of synthesis temperature.

It is C/S molar ratio

Please revise the whole manuscript again!

The reviewer cannot recommend this work for publication at this stage until the whole manuscript is carefully revised. The reviewer will leave the decision to the respected editor.

Response: The manuscript has been carefully revised according to the above suggestions, and the written expression and organization of whole manuscript also carefully revised and optimized again.

Reviewer 4 Report

Figure 1 and Figure 3i in this manuscript are the same as Figure 1 and Figure 3e in previous authors' research Journal of Materials research and technology 2021; 1 5: 4185 -4192. The authors should get permission from the journal to use these figures or should use another. 

Author Response

Figure 1 and Figure 3i in this manuscript are the same as Figure 1 and Figure 3e in previous authors' research Journal of Materials research and technology 2021; 1 5: 4185 -4192. The authors should get permission from the journal to use these figures or should use another. 

Response: Phase composition of calcium silicate slag and silica fume is analyzed through XRD again, and the results are shown as Figure.1 (a) and Figure.1(b) respectively. Figure. 3(I) is also replaced by another SEM image from the same synthesis condition.